# Phenotypic and molecular characterization of *Prototheca wickerhamii* from a Brazilian case of human systemic protothecosis

**Luciana Duarte-Silva[1], Raquel Vilela[1,2], Isabela A. Rodrigues[1], Vanessa C. R. Magalhães[3], Marcelo V. Caliari[4], Leonel Mendoza**  **[2]*, Adriana Oliveira Costa[1]**

**1** Departamento de Análises Clínicas e Toxicológicas, Faculdade de Farmácia, Universidade Federal de Minas Gerais, Brazil, **2** Microbiology, Genetics, and Immunology, Biomedical Laboratory Diagnostics, Michigan State University, East Lansing, Michigan, United States of America, **3** Hospital Eduardo de Menezes, Fundação Hospitalar, Belo Horizonte, Minas Gerais, Brazil, **4** Departamento de Patologia Geral, Instituto de Ciências Biológicas, Universidade Federal de Minas Gerais, Brazil

* mendoza9@msu.edu

## Abstract

The genus *Prototheca* (alga) comprises a unique group of achlorophyllic saprotrophic and mammalian pathogen species. Despite its rare occurrence in humans and animals, protothecosis is considered an emerging clinical entity with relevance in immunocompromised patients. In this study, the characterization of spherical structures with endospores recovered from a blood culture in an HIV patient was investigated using phenotypic and molecular methodologies. On 2% Sabouraud dextrose agar, the isolate displayed morphological and biochemical characteristics found on isolates identified as *Prototheca wickerhamii*. To validate these analyses, molecular phylogeny of the internal transcript space (ITS) partial gene confirmed the identity of the isolate as *P. wickerhamii*. This is the first case of systemic human protothecosis in Brazil. The present case of human *Prototheca* and those reported in the medical literature highlight the need for novel methodologies to identify pathogenic algae in the clinical laboratory, improving in this way the diagnosis and treatment of this group of neglected pathogens.

**Data Availability Statement:** Morphological and molecular minimal data set analyzed in this study is available within the manuscript itself. The DNA

## Author summary

Species of *Prototheca* are achlorophyllous algae widespread in aquatic and wet environments, exhibiting the surprising ability to act as a pathogen of mammals, including humans. With no tendency to self-resolve and the potential for fatal outcomes in immunocompromised patients, protothecosis has emerged as a concerning infection in the last decades. In this study, we identified a *Prototheca* isolate from a fatal case of systemic protothecosis, using traditional assays, and phylogenetic analysis. Traditional assays identified the pathogen as *Prototheca wickerhamii*, findings later confirmed by molecular methodologies. Because of its unique classification, clinical diagnosis and laboratory identification are challenging. For instance, the macro morphology of the colonies on culture mimics those in *Candida* species, a fact contributing to delay therapy. Therefore, the accurate

sequences and data in the manuscript have been deposited at the NCBI database under accession PP455257 and PQ369238. The isolate is in the process of submission to ATCC soon.

**Funding:** The author(s) received no specific funding for this work.

**Competing interests:** The authors have declared that no competing interests exist.

identification and characterization of clinical isolates using molecular methodologies may have a positive impact on the diagnosis and treatment of this unusual infection.

## Introduction

Human protothecosis is a rare algal infection caused by *Prototheca* species. These organisms are unicellular microbes lacking chloroplasts likely lost throughout their evolutionary history [1,2]. Phylogenetically, they are located basal to plants and the Straminipila fungi [3,4]. *Prototheca* species are heterotrophic microbes belong to the family Chlorellaceae, order Chlorellales, in the class Trebouxiophyceae, sharing phenotypic and phylogenetic similarities with the photosynthetic algae *Chlorella* and *Auxenochlorella* species [1,2,4]. *Prototheca* species survive in wet environments, including tap water and other natural sources of water, vegetables, saprophytic niches as sewage and animal manure, and in the intestinal tract of animals and their excrements [1,5,6].

*Prototheca* species contact the mammalian hosts through the traumatic inoculation of environmental propagules developing cutaneous, elbow joint (olecranon bursitis), nail (onycoprotothecosis), and rarely systemic infections [7]. They are considered opportunistic pathogens causing skin infections in immunocompromised hosts by disseminating through blood vessels developing systemic infections [1,8,9]. Among *Prototheca* spp. causing infections in mammalian hosts, *P. wickerhamii* is the most prevalent species. However, cases attributable to *P. zopfii* have also been diagnosed causing bovine mastitis [1,10]. In addition, the species *P. cutis* and *P. miyaji* were described as causative agents of cutaneous and disseminated protothecosis in humans [11,12]. The mechanisms underlying the pathogenesis and host-pathogen interaction in cases of protothecosis have yet to be investigated.

The diagnosis of protothecosis usually involves cytology, histopathology, and culture on Sabouraud dextrose agar. In this medium *Prototheca* spp. develop white yeast-like colonies. Microscopically, colonies on agar plates or in histologic preparations from clinical specimens, reveal spherical cells with morula-like features (sporangia) with numerous internal sporangiospores [9,13]. Worldwide, the number of well-documented protothecosis cases was estimated to be 211 [7]. However, the occurrence may be the top of iceberg considering the similarity of clinical features shared with other pathogens, including the pathogenic fungi [14]. Reported cases of protothecosis in Brazil involved the cutaneous form [14–16], onycoprotothecosis [17] and an olecranon bursitis case [18].

This study deals with the phenotypic and molecular characterization of a spherical pathogen with endospores isolated from a HIV patient with systemic infection admitted in a reference Hospital in Belo Horizonte, Minas Gerais, Brazil. The identification of the *P. wickerhamii* was the first report of this pathogen isolated from blood cultures in an unusual case of systemic infection in Brazil.

## Material and methods

### Ethics statement

The use of the culture for research, as well as the patient clinical data were approved by the Research Ethics Committee Federal University of Minas Gerais (UFMG—Number 2084026) and Hospital Eduardo de Menezes (Number 2110303).

### *Prototheca* isolate

The alga in this study was isolated in 2016 from blood culture in a 51 years-old male immuno-compromised HIV positive patient, at the Hospital Eduardo de Menezes, Belo Horizonte, Minas Gerais, Brazil. At admission, the patient presented an ulcerated 2.0 cm diameter skin lesion on the right foot, with symptomatology consistent with a systemic infection. Blood tests indicated anemia, leukopenia, and thrombocytopenia. A blood culture was requested, and after incubation, the sample was inoculated onto mycosel agar plates and subsequently transferred to 2% Sabouraud dextrose agar (SDA). During hospitalization, the patient received intravenous antibiotic therapy with oxacillin, but his general condition worsened, and he succumbed to his infection on the 24th day after admission.

### Phenotypic characterization

**Morphological observations.** Using a digital camera, the macroscopic aspect of the colonies was evaluated after 72h of incubation on SDA at 32˚C. Microscopic characterization of a single colony (2 to 5 mm in diameter) was carried out by diluting it in 1.0 mL of phosphate buffer saline (PBS pH 7.2). Slides prepared on lactophenol cotton blue (Sigma-Aldrich, St. Louis, MO, USA) were analyzed in an Axiolab microscope (Carl Zeiss, Oberkochen, Germany) and several images were captured with a coupled camera Samsung SDC-415 (Seoul, South Korea). The images were analyzed with the software (mi Carl Zeiss, Oberkochen, Germany) and the average diameter of 100 sporangia and 100 sporangiospores were estimated.

### Cell cycle observations in liquid medium

Ten microliters of *Prototheca* suspension in PBS were inoculated in three tubes containing 10.0 mL of PYEG liquid medium (peptone 20g/L KASVI, Brazil; Yeast Extract 2 g/L micro-MED/Laborchemiker, Brazil; Glucose 9 g/L Reagen Quimiobrás, Brazil) and then incubated at 37˚C for 72 hours. To separate larger sporangia from sporangiospores the above liquid medium was filtrated using an 8 μm filter (80 g/cm$^2$). The resulting filtrate containing sporangiospores was then dispensed in three tubes each containing 3.0 mL of the filtrate, and then incubated at 37˚C. For the following six days the initial inoculum (time 0h) was estimated in 100 stages/mL. *Prototheca* cell cycle stages (sporangia and sporangiospores) were quantified daily using a Neubauer chamber.

### Biochemical tests

Carbohydrate assimilation test was performed on a 24 wells plates containing agar peptone medium (10 g/L bacteriological peptone Kasvi 1.7% bacteriological agar, pH 7, 20 mg/L bromophenol blue) and each carbohydrate at 40% glucose (Reagen Quimiobrás, Brazil), fructose, galactose (BD Difco, Le Pont de Claix, France), trehalose and mannitol (Synth, São Paulo, Brazil). A suspension of *Prototheca* cells was prepared by dilution of a single colony from 72h-old culture in 1.0 mL of PBS. Ten microliters were inoculated in each well using striation technique in duplicate. The test was also performed by pour plate technique, in which *Prototheca* cells (500 μL) were added to 2 mL of the carbohydrate medium solubilized at 45˚C. The suspension was quickly homogenized and distributed in each well. The plates were incubated at 32˚C and evaluated daily for 4 days.

Sensitivity assay was performed using 10, 50 and 150 μg/mL concentrations of clotrimazole (Sigma-Aldrich, St. Louis, MO, USA) on Sabouraud dextrose agar medium. *Prototheca* suspension prepared from a colony 72h-old culture diluted in PBS pH 7.2, inoculated by striation on the culture plates in duplicate and incubated at 32˚C for 96 hours.

### DNA isolation, Sequencing, and phylogenetic analysis

**DNA extraction.** Genomic DNA was extracted using Sabouraud dextrose broth cultures incubated at 37˚C while being rotated at 150 rpm for 72 hours. After incubation, the culture was centrifuged, and the pellet was disrupted using Fungi/Yeast Genomic DNA isolation Kit protocol (Norgen Biotek Corp. Thorold, ON, Canada). The resulting mixture was treated with sodium dodecyl sulfate and digested with RNase A and protein K (Qiagen, Germantown, MD, USA) at 60˚C for 1 h. The DNA was extracted with phenol/chloroform, and the resulting DNA dissolved in nuclease free sterile distilled water and storage at − 80˚C. PCR reactions were performed with primers targeting *P. wickerhamii*'s 5.8S conserved region Proto-F1 5'ACTGCGAGACGTAGTGTG3' and Proto-R 5'TCAGGTCGCCACATGTGC3', and the primers used by Jagielski et al. [19] to amplify *Cytochrome oxidase* (*cytb*) partial gene. Ampli-Taq-Gold polymerase (Applied BioSystems, Branchburg, New Jersey, USA) was used in 25 μl volume reactions. The reactions were conducted under the following conditions: 95˚C for 10 min; 40 cycles consisting of 1 min at 95˚C, 2 min at 60˚C, and 3 min at 72˚C; final extension at 72˚C for 10 min. The amplicons were purified with PureLink Purification Kit (ThermoFisher Scientific, Carlsbad CA, USA) and then sequenced in both directions with the same primers using BigDye terminator chemistry in an ABI Prism 310 genetic analyzer (Perkin-Elmer Foster City, Cal.).

### Phylogenetic analysis

The sequenced amplicon was first investigated using Basic Local Alignment Search Tool (BLAST) and then aligned with homologous DNA sequences fetched from the National Center for Biotechnology Information (NCBI) in MEGA X software [20]. The DNA sequence was aligned using MEGA X software, inspected by eye and then used for phylogenetic analysis. Evolutionary history was inferred by using the Maximum Likelihood method and Kimura 2-parameter model [21]. In this analysis the tree with the highest log likelihood was selected. The analysis involved 42 nucleotide sequences. All positions with less than 95% site coverage were eliminated, i.e., fewer than 5% alignment gaps, missing data, and ambiguous bases were allowed at any position (partial deletion option). Evolutionary analyses were conducted in MEGA X [20].

## Results

On SDA after 72h of incubation at 32˚C the investigated isolate showed numerous opaque cream-white, yeast-like colonies with corrugated edges and central elevations (Fig 1A). Micro-scopically, the colonies comprise large spherical structures (sporangia) displaying morula-like features and holding numerous small endogenous spherical spores (sporangiospores). These structures were surrounded by numerous immature cells without endo-sporangiospores all features in common with *Prototheca* species (Fig 1B–1D).

Size measuring indicated that large sporangia have an average of 8.09 + 0.93 μm in diameter, whereas the sporangiospores measured 3.84 ± 0.57 μm in diameter (Fig 2A). In the liquid medium PYEG, sporangiospores increased exponentially for 96 hs, while sporangia number slightly increased after 120hs (Fig 2B).

Using color change (blue to yellow) as indicator of sugar hydrolysis, the biochemical assays showed the investigated isolate hydrolyzed glucose, trehalose, and fructose, but not galactose and mannitol. Change pattern after 96h of incubation was better visualized using pour plate compared to striation technique, particularly for glucose (Fig 3). Regarding the sensitivity to clotrimazole, no colony developed in the medium after 96 hours of incubation, suggesting *in vitro* the isolate is susceptible to clotrimazole. Overall, the biochemical assays showed sugar

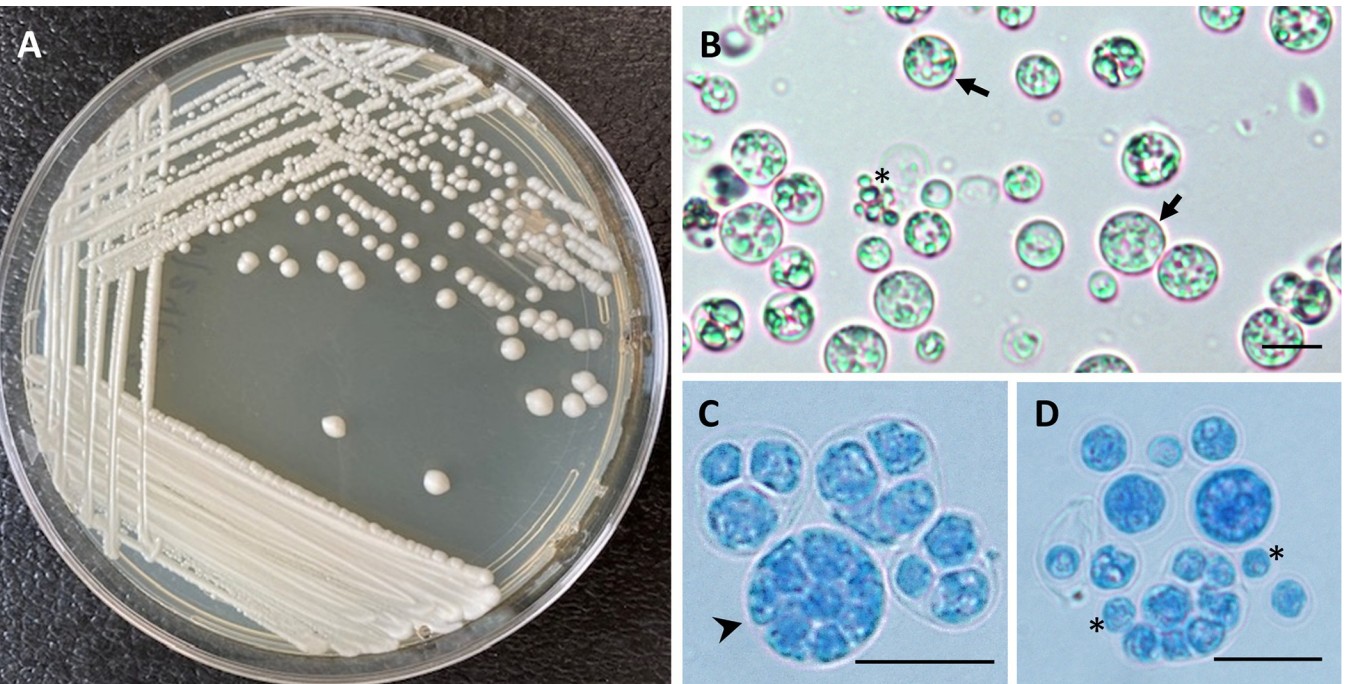

**Fig 1. Morphological characterization of *Prototheca* sp.** A. Macroscopic aspect of colonies cultured in SDA for 72 hours at 32°C, showing corrugated edges with central elevation of the colonies and their cream-white pigmentation. B. Wet preparation from SDA colonies showing microscopic characteristics of spherical sporangia (arrows) and sporangiospores (asterisk). Bar: 10 μm. C, D. Detail of morula-like sporangia (arrowhead) and sporangiospores (asterisk) in lactophenol cotton blue stain. Bar: 10 μm.

assimilation and clotrimazole sensitivity patterns were compatible with that of expected for *P. wickerhamii* (Table 1).

To validate the phenotypic and biochemical data of the investigated isolate, BLAST and phylogenetic analysis were performed with partial ITS DNA sequence (accession number PP455257) and *cytb* partial gene (accession number PQ369238). BLAST analysis of the ITS

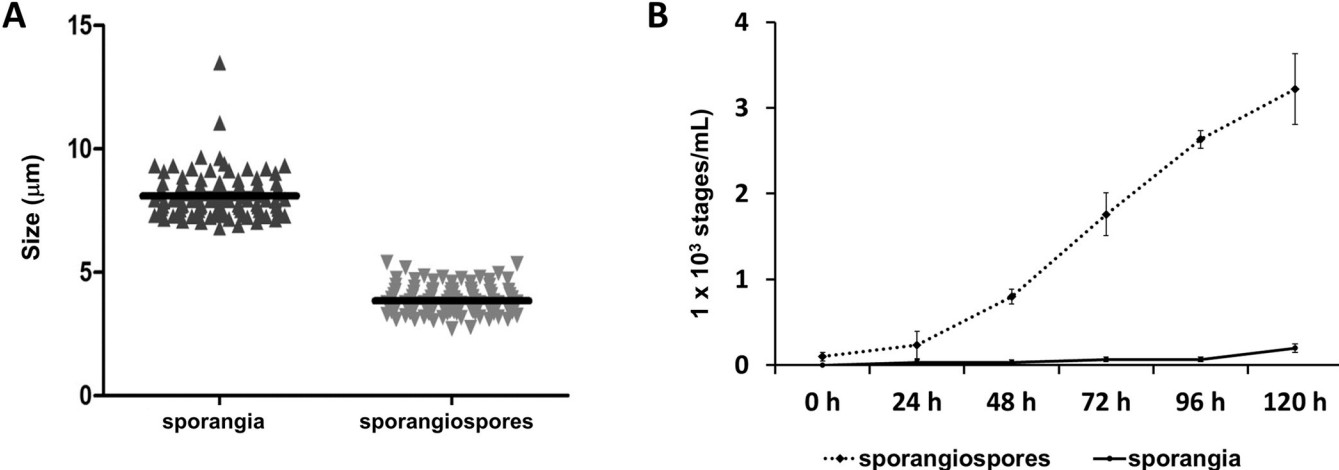

**Fig 2. Morphometric and growth evaluation of *Prototheca* stages.** A. *Prototheca* stages in SDA have a size of 8.09 + 0.93 μm for sporangia and 3.84 ± 0.57 μm for sporangiospores. Digital images analyzed by KS400 software (Carl Zeiss, Oberkochen, Germany). B. Cell cycle pattern in PYEG liquid medium after an inoculum of 100 sporangiospores/mL.

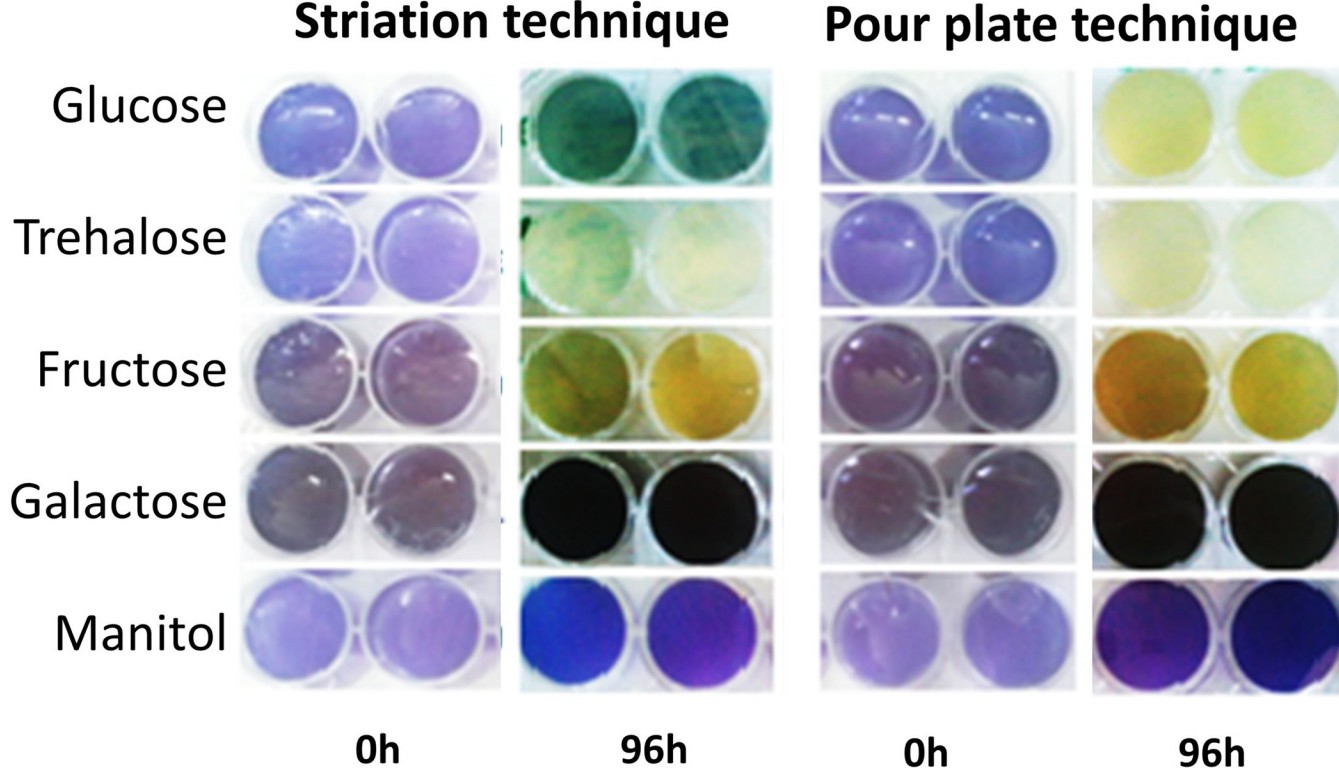

**Fig 3. Sugar assimilation assay using striation and pour plate techniques.** The assays were performed in agar peptone medium (10 g/L bacteriological peptone, 1.7% bacteriological agar, 20mg/L bromophenol blue, pH 7.0) containing 40% of each carbohydrate. Image obtained after 96h of incubation at 32˚C.

and the *cytb* partial DNA sequences showed 99.79% and 100% identity respectively, with *P. wickerhamii* DNA sequences. Phylogenetic analysis of the ITS DNA sequences from 36 *Prototheca* DNA sequences (including the Brazilian DNA sequence), and six *Auxenochlorella protothecoides* (*Chlorella protothecoides*), the Brazilian DNA sequence clustered within the 29 *P. wickerhamii* DNA sequences and away from other *Prototheca* and *Auxenochlorella* species used in this analysis (Fig 4).

## Discussion

Prototheocosis is a rare infection of worldwide distribution [1,9,23,24]. It mostly occurs as single isolated cases primarily affecting the skin of the infected hosts [1,5,15–18]. Disseminated

**Table 1. Identification of *Prototheca* species by biochemical assays including the investigated isolate in this study.**

| Species | Carbohydrate assimilation [a] | | | | | Clotrimazol sensitivity[b] |
|---|---|---|---|---|---|---|
| | **Fructose** | **Galactose** | **Glucose** | **Mannitol** | **Trehalose** | |
| *P. moriformis* | + | - | + | - | - | NT |
| *P. stagnora* | + | + | + | - | - | Susceptible |
| *P. ulmea* | - | - | + | - | - | NT |
| *P. wickerhamii* | + | V | + | - | + | Susceptible |
| *P. zopfii* | + | - | + | - | - | Resistant |
| Isolate from this study | + | - | + | - | + | Susceptible |

a. Endo et al., (2010) [21]

b. Casal and Gutierrez, 1983 [22]; V: variable; NT: not tested.

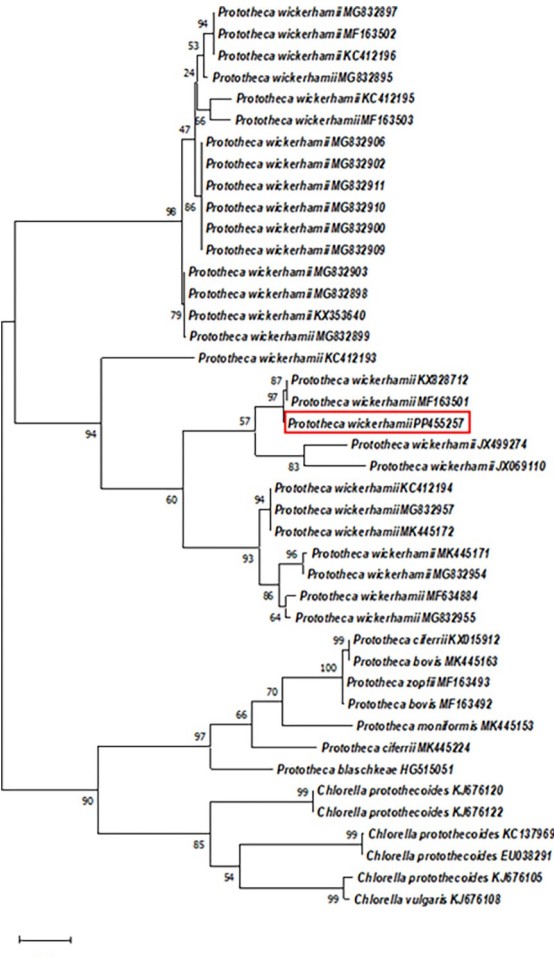

**Fig 4. Unrooted phylogenetic tree of partial ITS1, 5.8S, and ITS2 DNA sequences from various *Prototheca* and *Chlorella* (*Auxenochlorella*) species (accessions numbers shown on the tree).** The Brazilian isolated clustered with other *P. wickerhamii* DNA sequences (red rectangle). Evolutionary history was inferred by using the Maximum Likelihood method and Kimura 2-parameter model [21]. The tree with the highest log likelihood (-940.82) is shown. The percentage of trees in which the associated taxa clustered together is shown next to the branches. Initial tree(s) for the heuristic search were obtained automatically by applying Neighbor-Join and BioNJ algorithms to a matrix of pairwise distances estimated using the Maximum Composite Likelihood (MCL) approach, and then selecting the topology with superior log likelihood value. A discrete Gamma distribution was used to model evolutionary rate differences among sites (6 categories [+G, parameter = 2.5023]). The tree is drawn to scale, with branch lengths measured in the number of substitutions per site.

cases of protothecosis, like the one described in this study, are unusual and occur predominantly in immunocompromised individuals [12,24–31].

The incidence of protothecosis increased over the last 10 years, with more than 200 cases reported to date [6,7]. Systemic protothecosis accounts for 37 cases until 2019, none of them in South America [6]. This is the first systemic case recorded in Brazil. The patient was admitted to the Hospital with cutaneous lesions in his lower limbs, suggesting the origin of his systemic infection was skin wounds. This is supported by the many cases of protothecosis involving traumatic implantation of this pathogenic alga from environmental sources [1]. Therefore, the cutaneous inoculation of *Prototheca* spp. seems to be the typical epidemiological feature among infected hosts. However, colonization of the gut followed by translocation could also be considered in cases of systemic dissemination. The latter endogenous via was recently

suggested to explain the origin of an outbreak of *P. wickerhamii*, causing algaemia and sepsis in 12 neutropenic oncology patients receiving chemotherapy in India [8].

The characterization of the *Prototheca* isolate in this study included macro- and microscopic morphological features of the colonies as well as phylogenetic analysis [1,31]. Our study showed that the 5.8S ITS region of *P. wickerhamii* together with the *cytb* partial gene [19] are good markers to ID this pathogen from clinical isolates. The cell cycle and the size estimation, and their sugar hydrolysis reaction were consistent with *P. wickerhamii* species, displaying sporangia and sporangiospores smaller than other mammalian pathogenic species including *P. zopfii* [1]. The Brazilian *Prototheca* isolate hydrolyzed glucose, trehalose, and fructose, but not galactose. Trehalose assimilation by *P. whickerhamii* is the main difference with *P. zopfii* that does not hydrolyze this sugar. Galactose assimilation is variable in *P. wickerhamii* and is always negative for *P. zopfii* [1,22]. In the biochemical assay, the pour plate technique performed better than striation for sugar assimilation assays. The Brazilian isolate was susceptible to clotrimazole, a finding that further separate this isolate from that of *P. zopfii* resistant to clotrimazole [23].

Unfortunately, before antifungal treatment could be implemented, the patient developed systemic shock and subsequently succumbed to his infection. This case and others with fatal outcome [26,27] supporting the concept that the diagnosis and therapeutic management of systemic protothecosis are challenging. In summary, the isolation of *P. wickwerhamii* in this study was the first report in Brazil from a human case of protothecosis developing algaemia. The pathogenesis of algal microbes remains poorly understood, and thus the disease may be underdiagnosed due to the lack of diagnostic tools to properly identify this pathogen. Therefore, the use of molecular methodologies, such as the one in this study, was key for the accurate identification of the pathogen. The use of this technology may have implications for the diagnosis of future cases and the appropriate management of this unique neglected pathogen.

## Acknowledgments

The authors thank the personnel at the Hospital Eduardo de Menezes, Belo Horizonte, Minas Gerais, Brazil for agreeing to provide a subculture of *Prototheca*.

## Author Contributions

**Conceptualization:** Vanessa C. R. Magalhães, Marcelo V. Caliari, Leonel Mendoza, Adriana Oliveira Costa.

**Data curation:** Raquel Vilela, Isabela A. Rodrigues, Leonel Mendoza.

**Formal analysis:** Luciana Duarte-Silva, Raquel Vilela, Vanessa C. R. Magalhães.

**Investigation:** Raquel Vilela, Leonel Mendoza, Adriana Oliveira Costa.

**Methodology:** Adriana Oliveira Costa.

**Project administration:** Luciana Duarte-Silva, Adriana Oliveira Costa.

**Supervision:** Raquel Vilela, Adriana Oliveira Costa.

**Validation:** Raquel Vilela, Leonel Mendoza.

**Writing – original draft:** Luciana Duarte-Silva, Raquel Vilela, Isabela A. Rodrigues, Vanessa C. R. Magalhães, Marcelo V. Caliari, Leonel Mendoza, Adriana Oliveira Costa.

**Writing – review & editing:** Luciana Duarte-Silva, Raquel Vilela, Isabela A. Rodrigues, Vanessa C. R. Magalhães, Marcelo V. Caliari, Leonel Mendoza, Adriana Oliveira Costa.

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
