## [Decision Letter · Decision Letter 0]

23 Apr 2024

Dear Dr. Mendoza,

Thank you very much for submitting your manuscript "Phenotypic and molecular characterization of Prototheca wickerhamii from a Brazilian case of human systemic protothecosis" for consideration at PLOS Neglected Tropical Diseases. As with all papers reviewed by the journal, your manuscript was reviewed by members of the editorial board and by several independent reviewers. In light of the reviews (below this email), we would like to invite the resubmission of a significantly-revised version that takes into account the reviewers' comments. 

We cannot make any decision about publication until we have seen the revised manuscript and your response to the reviewers' comments. Your revised manuscript is also likely to be sent to reviewers for further evaluation.

Sincerely,

Joshua Nosanchuk, MD

Section Editor

Joshua Nosanchuk

Section Editor

Reviewer's Responses to Questions

**Key Review Criteria Required for Acceptance?**

**Methods**

-Are the objectives of the study clearly articulated with a clear testable hypothesis stated?

-Is the study design appropriate to address the stated objectives?

-Is the population clearly described and appropriate for the hypothesis being tested?

-Is the sample size sufficient to ensure adequate power to address the hypothesis being tested?

-Were correct statistical analysis used to support conclusions?

-Are there concerns about ethical or regulatory requirements being met?

Reviewer #1: (No Response)

Reviewer #2: The objective of the study is well defined. The methodological approach is the most appropriate. The methodology is described in detail, which allows them to be reproduced without problem.

**Results**

-Does the analysis presented match the analysis plan?

-Are the results clearly and completely presented?

-Are the figures (Tables, Images) of sufficient quality for clarity?

Reviewer #1: (No Response)

Reviewer #2: The results are presented clearly. The molecular data (sequence) of the identified isolate is deposited in GenBank.In Table 1, the result of the sensitivity to clotrimazole of the studied isolate should be “susceptible” not “+” to facilitate comparison between Prototheca species.

It is necessary to improve the visual quality of figure 4, the text is not visible well

**Conclusions**

-Are the conclusions supported by the data presented?

-Are the limitations of analysis clearly described?

-Do the authors discuss how these data can be helpful to advance our understanding of the topic under study?

-Is public health relevance addressed?

Reviewer #1: (No Response)

Reviewer #2: The conclusion of the work is well supported. The manuscript does not present limitations of the study. The authors mention the importance of phenotypic and genotypic identification of Proteotheca to administer adequate treatment to patients with protothecosis.

**Editorial and Data Presentation Modifications?**

Reviewer #1: (No Response)

Reviewer #2: In Table 1, the result of the sensitivity to clotrimazole of the studied isolate should be “susceptible” not “+” to facilitate comparison between Prototheca species.

The authors mention that “Phylogenetic analysis of DNA sequences from 36 Prototheca

species (including the Brazilian DNA sequence), and six Auxenochlorella protothecoides

(Chlorella protothecoides)”, but I do not see sequences of 36 species of Prototheca, I see 36 sequences of at least 6 species of Prototheca

**Summary and General Comments**

Reviewer #1: This article ignores a bulk of recent literature on Prototheca algae, without which any discussion and/or experimental study (even of a clinical case report) are pointless. First of all the authors keep silent about Prototheca taxonomy. The unawareness of valid Prototheca taxonomy reverberates also in the Discussion, where species names no longer valid are used (e.g. P. zopfii). Nothing is mentioned about how many species are currently recognized, and which of them are of clinical relevance. There is no word on very recent findings regarding early Prototheca genomic studies.

When planning the identification algorithm, the authors completely disregarded the current standard in the molecular typing (speciation) of Prototheca spp., which involved the partial cytb gene. (Based on this marker the current Prototheca classification system has been built). The ITS locus, used by the authors is not a good marker for Prototheca spp. Another important drawback is that the authors did not provide details regarding culture collection, in which their strain should be deposited.

Furhtermore, a lot of important studies concerning new treatment alternatives for Prototheca infections are missing. Even the latest and the only such paper on in vitro drug susceptible of human Prototheca isolates has been overlooked. 

Finally, the language of the article needs to be radically improved.

In conclusion, I did not want to reject the paper right away, as I acknowledge the emerging problem that protothecosis has recently become, but the authors have to re-conceive their study, so that it is based on the knowledge and methodologies, currently valid or in use in the field of Prototheca research. Below are only some papers, which MUST be taken into account when updating the facts on Prototheca and protothecosis, re-designing the experimental part (identification of the pathogen), and discussing the results obtained.

https://www.sciencedirect.com/science/article/abs/pii/S2211926419303509

https://pubmed.ncbi.nlm.nih.gov/36300932/

https://pubmed.ncbi.nlm.nih.gov/33750287/

https://pubmed.ncbi.nlm.nih.gov/36943065/

https://pubmed.ncbi.nlm.nih.gov/30068534/

https://pubmed.ncbi.nlm.nih.gov/34791104/

https://pubmed.ncbi.nlm.nih.gov/29790400/

https://pubmed.ncbi.nlm.nih.gov/29725298/

https://pubmed.ncbi.nlm.nih.gov/30965146/

Reviewer #2: The manuscript “Phenotypic and molecular characterization of Prototheca wickerhamii from a Brazilian case of human systemic protothecosis” presents the identification of a Prototheca isolate from a fatal case of systemic protothecosis, using traditional assays, and phylogenetic analysis. This isolate represents the first case of systemic protothecosis in Brazil. This report is relevant in its field of study, since protothecosis is a rare disease, and identifying the pathogen is a challenge. In this study, this challenge is addressed using phenotypic and molecular tests. The methodological approach is the most appropriate. The methodology is described in detail, which allows them to be reproduced without problem. The results are presented clearly. The molecular data (sequence) of the identified isolate is deposited in GenBank.

My main criticism is that the authors do not mention what treatment the patient received. However, do they perform sensitivity tests for clotrimazole? Was it clotrimazole that the patient received?

PLOS authors have the option to publish the peer review history of their article (what does this mean?). If published, this will include your full peer review and any attached files.

Reviewer #1: No

Reviewer #2: No
---

## [Decision Letter · Decision Letter 1]

3 Sep 2024

Dear Dr. Mendoza,

Thank you very much for submitting your manuscript "Phenotypic and molecular characterization of Prototheca wickerhamii from a Brazilian case of human systemic protothecosis" for consideration at PLOS Neglected Tropical Diseases. As with all papers reviewed by the journal, your manuscript was reviewed by members of the editorial board and by several independent reviewers. In light of the reviews (below this email), we would like to invite the resubmission of a significantly-revised version that takes into account the reviewers' comments. I want to clearly acknowledge that one reviewer is highly positive of this work; however, the other reviewer is extremely critical of the response provided. Given the dichotomy of views, we are providing you with the opportunity to revise this work and we will coordinate a review with a new reviewer of the work. We acknowledge that this is a challenge for you, but it is consistent with our required rigor of review for the journal. 

We cannot make any decision about publication until we have seen the revised manuscript and your response to the reviewers' comments. Your revised manuscript is also likely to be sent to reviewers for further evaluation.

Sincerely,

Joshua Nosanchuk, MD

Section Editor

Reviewer's Responses to Questions

**Key Review Criteria Required for Acceptance?**

**Methods**

-Are the objectives of the study clearly articulated with a clear testable hypothesis stated?

-Is the study design appropriate to address the stated objectives?

-Is the population clearly described and appropriate for the hypothesis being tested?

-Is the sample size sufficient to ensure adequate power to address the hypothesis being tested?

-Were correct statistical analysis used to support conclusions?

-Are there concerns about ethical or regulatory requirements being met?

Reviewer #1: Please read below.

Reviewer #2: The objective of the study is clear, and the design is consistent with the objective. The methodology used is appropriate.

**Results**

-Does the analysis presented match the analysis plan?

-Are the results clearly and completely presented?

-Are the figures (Tables, Images) of sufficient quality for clarity?

Reviewer #1: Please read below.

Reviewer #2: The analysis presented for the identification of the isolate is adequately and clearly detailed. 

The figures are of sufficient quality and perfectly illustrate what is described in the text. However, it would be ideal if the quality of Figures 4 could be improved.

**Conclusions**

-Are the conclusions supported by the data presented?

-Are the limitations of analysis clearly described?

-Do the authors discuss how these data can be helpful to advance our understanding of the topic under study?

-Is public health relevance addressed?

Reviewer #1: Please read below.

Reviewer #2: The conclusions are supported by the results presented. The authors clearly state the relevance of the findings to public health.

**Editorial and Data Presentation Modifications?**

Reviewer #1: Please read below.

Reviewer #2: I suggest taking care of the quality of figure 4.

**Summary and General Comments**

Reviewer #1: This is the second time I have read this paper. Sadly, the authors did not provide answers to my comments or the answers were not satisfactory/convincing to me.

Again I cannot agree with the authors who insistently claim that the ITS is a good marker for Prototheca identification. Obviously, it is not. I am quite surprised that the authors, as they say, could not find it in the literature. One clear example is inability of ITS to discriminate between P. wickerhamii and P. xanthoriae (read more on this here: https://www.sciencedirect.com/science/article/abs/pii/S2211926419303509 ). One may say that the latter is if no clinical significance. Note, however, that both P. bovis and P. ciferii have long been considered as being associated with bovine mastitis only, while they have been recently demonstrated as human pathogens (!).

Furthermore, referring to databases abundant with ITS sequences is not a good point, since many of these sequences had not been properly checked/processed etc. This is in fact a problem with many Prototheca sequences which do not undergo extensive curation prior to submission. This problem has been overcome with the development of the Prototheca-ID database, which provides an accurate identification with easily produced phylograms.

The authors clearly did not look watchfully into the problem of Prototheca taxonomy, as they have introduced much confusion over the species names (e.g. P. ciferrii versus P. zopfii; P. ciferrii replaced the former genotype P. zopfii gen. 1; P. zopfii does existst as a separate species, but has nothing to do with the previously known P. zopfii genotypes 1 & 2). And although I understand that the Prototheca taxonomy per se is not a key objective of the study, it MUST NOT be approached with such a desinvolture.

Overall, I do not understand why the authors did not perform strain identification based on cytb gene, as requested. This would have cost almost nothing (both in terms of money and time. Technically typing using partial cytb gene is by no means more difficult than typing with other markers, including ITS). Finally, contrary to author’s claims, the cytb gene-based typing is not restricted to phylogenetic studies, as it was originally designed for a clinical setting as well.

To conclude, the authors uphold their claims, which I cannot agree with. The message conveyed by this study perpetuates the erroneous, in my opinion, view of the appropriateness of ITS as a marker for identification of Prototheca species.

Reviewer #2: The authors have heeded the reviewers' suggestions and the current version has improved considerably.

PLOS authors have the option to publish the peer review history of their article (what does this mean?). If published, this will include your full peer review and any attached files.

Reviewer #1: No

Reviewer #2: No
---

## [Editor Report · Decision Letter 2]

3 Oct 2024

Dear Dr. Mendoza,

Thank you for your clear and thoughtful response to the challenging review. We are pleased to inform you that your manuscript 'Phenotypic and molecular characterization of Prototheca wickerhamii from a Brazilian case of human systemic protothecosis' has been provisionally accepted for publication in PLOS Neglected Tropical Diseases.

Best regards,

Joshua Nosanchuk, MD

Section Editor

Joshua Nosanchuk

Section Editor

---

## [Editor Report · Acceptance letter]

27 Oct 2024

Dear Dr. Mendoza,

We are delighted to inform you that your manuscript, "Phenotypic and molecular characterization of Prototheca wickerhamii from a Brazilian case of human systemic protothecosis," has been formally accepted for publication in PLOS Neglected Tropical Diseases.

Best regards,

Shaden Kamhawi

co-Editor-in-Chief

Paul Brindley

co-Editor-in-Chief
